# Overcoming Microenvironment-Mediated Chemoprotection through Stromal Galectin-3 Inhibition in Acute Lymphoblastic Leukemia

**DOI:** 10.3390/ijms222212167

**Published:** 2021-11-10

**Authors:** Somayeh S. Tarighat, Fei Fei, Eun Ji Joo, Hisham Abdel-Azim, Lu Yang, Huimin Geng, Khuchtumur Bum-Erdene, I. Darren Grice, Mark von Itzstein, Helen Blanchard, Nora Heisterkamp

**Affiliations:** 1Division of Hematology/Oncology and Bone Marrow Transplant, The Saban Research Institute of Children’s Hospital Los Angeles, Los Angeles, CA 90027, USA; somayeh.tarighat@gmail.com (S.S.T.); feifeilike@gmail.com (F.F.); astrojej@gmail.com (E.J.J.); HAbdelAzim@chla.usc.edu (H.A.-A.); 2Department of Systems Biology, Beckman Research Institute, City of Hope, Monrovia, CA 91016, USA; luyang@coh.org; 3Department of Laboratory Medicine, University of California, San Francisco, CA 94143, USA; Huimin.Geng@ucsf.edu; 4Institute for Glycomics, Griffith University, Gold Coast, Southport, QLD 4222, Australia; bumerdek@iu.edu (K.B.-E.); d.grice@griffith.edu.au (I.D.G.); m.vonitzstein@griffith.edu.au (M.v.I.); hblancha@uow.edu.au (H.B.); 5School of Medical Science, Griffith University, Gold Coast, Southport, QLD 4222, Australia; 6School of Chemistry and Molecular Bioscience and Molecular Horizons, University of Wollongong, Wollongong, NSW 2522, Australia; 7Illawarra Health & Medical Research Institute, Wollongong, NSW 2522, Australia

**Keywords:** B-cell precursor ALL, galectin-3, *lgals3*, galectin, microenvironment, adhesion, migration, drug resistance, glycomimetic, carbohydrate-based galectin-3 inhibitor, monosaccharide, taloside

## Abstract

Environmentally-mediated drug resistance in B-cell precursor acute lymphoblastic leukemia (BCP-ALL) significantly contributes to relapse. Stromal cells in the bone marrow environment protect leukemia cells by secretion of chemokines as cues for BCP-ALL migration towards, and adhesion to, stroma. Stromal cells and BCP-ALL cells communicate through stromal galectin-3. Here, we investigated the significance of stromal galectin-3 to BCP-ALL cells. We used CRISPR/Cas9 genome editing to ablate galectin-3 in stromal cells and found that galectin-3 is dispensable for steady-state BCP-ALL proliferation and viability. However, efficient leukemia migration and adhesion to stromal cells are significantly dependent on stromal galectin-3. Importantly, the loss of stromal galectin-3 production sensitized BCP-ALL cells to conventional chemotherapy. We therefore tested novel carbohydrate-based small molecule compounds (Cpd14 and Cpd17) with high specificity for galectin-3. Consistent with results obtained using galectin-3-knockout stromal cells, treatment of stromal-BCP-ALL co-cultures inhibited BCP-ALL migration and adhesion. Moreover, these compounds induced anti-leukemic responses in BCP-ALL cells, including a dose-dependent reduction of viability and proliferation, the induction of apoptosis and, importantly, the inhibition of drug resistance. Collectively, these findings indicate galectin-3 regulates BCP-ALL cell responses to chemotherapy through the interactions between leukemia cells and the stroma, and show that a combination of galectin-3 inhibition with conventional drugs can sensitize the leukemia cells to chemotherapy.

## 1. Introduction

Relapse is a main cause of treatment failure for patients with B-cell precursor acute lymphoblastic leukemia (BCP-ALL) [1]. The tumor microenvironment is a major contributing factor to relapse because it regulates the migration, survival, proliferation and response to drug treatment in BCP-ALL cells [2,3]. Notably, leukemia cell movement and adhesion to the right niche depends on the ability of these cells to migrate towards chemotactic cues such as SDF-1 produced by stromal cells [4]. Interfering with leukemia-microenvironment interactions has been proven effective in mobilizing leukemia cells away from their protective microenvironment, thus making them more accessible to standard chemotherapy [5,6,7].

We and others have modeled the interaction between BCP-ALL cells and stroma in an ex vivo tissue co-culture model with OP9 bone marrow stromal cells to identify interactions that promote leukemia cell survival. Mesenchymal bone marrow stromal cells synthesize and secrete particularly high amounts of galectin-3 (Gal3), a carbohydrate-binding protein with immunomodulatory activity (for example, [8,9,10]). We previously reported that Gal3 acts as a communicator between BCP-ALL cells and the stroma: it not only binds to the cell surface of BCP-ALL cells but is also actively internalized by them [11,12]. Gal3 has a range of glycoconjugate ligands on the surface of cells, but intracellularly it also binds proteins [13] and regulates a variety of functions, including growth and mRNA splicing [14,15,16].

Although we previously showed that Gal3 protein synthesized endogenously in BCP-ALL cells promotes their survival [11], the physiological effect of Gal3 produced by the tumor microenvironment on BCP-ALL cells was unknown. Here, we have approached this problem in two ways. Using Cas9/CRISPR we have knocked out Gal3 in bone marrow stromal cells to determine if any BCP-ALL functions are regulated by stromal-produced Gal3. We also used novel small molecule monosaccharide-based carbohydrate mimetics to examine the effect of drug-mediated Gal3 inhibition on BCP-ALL physiology. We conclude that Gal3 is a valid target for enhancing the effects of standard chemotherapy by interfering with the communication between BCP-ALL and stromal cells.

## 2. Materials and Methods

### 2.1. Cells and Cell Culture

The murine OP9 bone marrow-derived stromal cell line (CRL-2749) was obtained from the American Type Culture Collection (Manassas, VA, USA). The patient derived TXL2, BLQ1 (Ph-positive) and ICN3, ICN13, US7, ICN06, LAX39 and LAX40 (Ph-negative) pre-B acute lymphoblastic leukemias were described previously [17,18]. BCP-ALL cells were grown on confluent irradiated or 10 μg/mL mitomycin-treated OP9 stromal cells using αMEM medium supplemented with 20% fetal bovine serum, 1% L-glutamine and 1% penicillin/streptomycin (Life Technologies, Grand Island, NY, USA).

To isolate normal human BM MSC, screens used to filter products intended for bone marrow transplants and which are otherwise discarded were rinsed with 15 mL DOM media to dislodge cells. After removal of debris via a 40 μm cell strainer and centrifugation, cells were suspended in either αMEM + 20% FBS or in DOM medium [IMDM + 12.5% horse serum + 12.5% FBS]. 1% L-glu, 0.5% P/S and 0.1% β-mercaptoethanol were added and cells were cultured at normoxia on 10-cm dishes. Cells were passaged by trypsinization when confluent. MSC were defined as a population that is CD45^neg^ and positive for SSEA-4 [19] and CD271/LNGFR [20].

Primary BCP-ALL and normal bone marrow mononuclear cells were purified using Ficoll (#17144002, GE Healthcare, Pittsburgh, PA, USA). Human specimen collection protocols were reviewed and approved by Children’s Hospital Los Angeles Institution Review Board (IRB) [Committee on Clinical Investigations] (CCI). Collections were in compliance with ethical practices and IRB approvals.

The synthesis and characterization of the taloside-based compounds 14 (Cpd 14; designated compound 1 in [21]) and 17 (Cpd 17; Bum-Erdene, manuscript submitted) are described in detail elsewhere. The compounds were dissolved in DMSO and stored at −80 °C. Aliquots were not used more than three times. Nilotinib (AMN107) (Novartis, Basel, Switzerland), and vincristine (Hospira Worldwide Inc., Lake Forest, IL, USA) (Sigma-Aldrich, St. Louis, MO, USA) were dissolved in DMSO and stored at −20 °C.

### 2.2. Fluorescence Activated Cell Sorting (FACS)

For cell surface staining, cells were washed 1× with PBS and passed through a 40 μm filter (BD Biosciences, San Jose, CA, USA) and blocked for 20 min at room temperature with FcR blocking reagent (Miltenyi Biotech, San Diego, CA, USA). Cells were then stained with 1 µg antibody per 1 × 10^6^ cells for 15 min at room temperature in the dark. After 2× washing with PBS, cells were resuspended in 200 μL FACS buffer (1× PBS, 1% FBS, 1 mM EDTA, 25 mM HEPES pH 7) and analyzed on a BD Accuri C6 cytometer (BD Biosciences, San Jose, CA, USA). As a control, cells were stained with a fluorophore-conjugated mouse IgG control. For intracellular staining, cells were first fixed with IC Fixation buffer (eBioscience, San Diego, CA, USA) for 30 min on ice, and permeabilized with Perm/Wash buffer (#554723, BD Biosciences) before blocking and staining.

### 2.3. Cell Cycle and Apoptosis

For cell cycle analysis, cells were fixed and permeabilized prior to resuspension in PI/RNase Staining buffer (BD Biosciences) for DNA content measurements and FACS analysis. For apoptosis detection, cells were resuspended in 200 μL 1× binding buffer with or without 5 μL Annexin V (BD Biosciences, Billerica, MA, USA). After 15 min incubation, cells were washed with PBS, resuspended in 500 μL FACS buffer and 5 μL propidium iodide (BD Biosciences) and analyzed by FACS. PHA-739358 (Danusertib) is a pan-Aurora kinase inhibitor with activity against all Aurora kinase family members (A, B and C). US7 was treated with 1 μM of PHA-739358 (24 h) as positive control for cell cycle analysis. Cells were fixed, permeabilized and stained with PI/RNase before FACS analysis. PHA-739358-treated cells present an increased G2/M-phase population. Data were analyzed using FCS Express 5 Plus Research Edition and FlowJo V10. BCP-ALL cells were harvested from the medium and/or were easily distinguished from OP9 cells-see Appendix A in [17].

### 2.4. Viability and Proliferation, Drug Treatment

Leukemia cells were cultured with an irradiated OP9 stroma layer in 24-well plates in complete αMEM in the presence of control DMSO or drugs (nilotinib, vincristine, Cpd 14 or Cpd17). Except where indicated otherwise, viable and total cell numbers were determined using Trypan blue exclusion. Growth curve assays were done by counting live cells using Trypan blue exclusion and an inverted microscope for >12 days after plating the cells, during which fresh media was supplied every two to three days. In long-term assays, fresh drugs were faded with the medium changes. Cpd14-induced inhibition of cell growth was also measured using a colorimetric AlamarBlue cell viability reagent (#PI88951, Thermo Fisher Scientific, Waltham, MA, USA). Cells (0.5–1 × 10^5^) were plated in 100 μL final volume in a 96-well plate in the presence of the different concentrations of Cpd14 for 24 h at 37 °C. Then 10 μL of the AlamarBlue reagent was added to each well. Within 5–6 h incubation at 37 °C, the fluorescence intensity at 535 and 590 nm was measured. Cell viability was calculated with respect to the control samples and reference background wavelength. At least three independent experiments were performed.

### 2.5. Adhesion

A 96-well plate was coated with 5 μg/mL fibronectin in PBS, overnight at 4 °C for 20 min at room temperature. BCP-ALL wells were washed with 0.1% BSA in PBS prior to blocking with 2% BSA in PBS for 1 h at room temperature. Wells were then washed twice with 0.1% BSA in PBS and labeled with 5 µM Calcein AM (#C3100MP, Thermo Fisher Scientific, Waltham, MA, USA) for 30 min at 37 °C, then washed 2× with PBS prior to seeding at 5 × 10^4^ cells/well in αMEM base media with or without drugs for 30 min. β-Lactose (Sigma-Aldrich, St. Louis, MO, USA) was included as a positive control for galectin-3 binding inhibition. After 30 min, the plate was read at 485 nm on a Synergy HTX Multi-Mode Reader (BioTek, Winooski, VT, USA). The plate was washed two more times with 0.1% BSA in PBS to completely remove unbound cells and read again at 485 nm. The percentage of adherent cells was calculated using the following formula: [(RFU post wash-RFU background)/(RFU pre wash-RFU background)] × 100%. To measure US7 and TXL2 adhesion to OP9 EV and OP9-KO cells, we collected floating cells and, after trypsinization, OP9 and associated ALL leukemia cells. Trypan blue was used to identify living cells, and manual counting was used to identify BCP-ALL cells in the mixture samples.

### 2.6. Migration

Migration of BCP-ALL cells was also tested using a 24-well Transwell system (5 µm pore size) (#3421, Costar, Corning Incorporated, NY, USA). Migration assays were carried out in αMEM with 2% FBS. BCP-ALL cells (0.5–1 × 10^5^) were plated inside the upper mesh insert (100 µL volume) while either 200 ng/mL SDF-1α (#300-28A, PeproTech, Rocky Hill, NJ, USA) or a monolayer of OP9 cells (1–2 × 10^4^ cells/well) in the lower chamber (final 600 µL of media) served as a chemoattractant. The number of migrated BCP-ALL cells in the lower chamber was counted by Trypan blue exclusion, with migration to OP9 measured after overnight incubation, and migration to SDF-1α after 2–4 h incubation.

### 2.7. Plasmids and CRISPR/Cas9

Galectin-3 cDNA was subcloned into the pMIG-GFP retroviral expression vector (plasmid #9044, AddGene, Cambridge, MA, USA) and included an HA-tag. Stably transduced hGalectin-3 expressing OP9 cells were generated by infecting cells with retroviral particles and FACS sorting of GFP expressing (positively transduced) cells [11]. Western blotting after cell expansion was used to confirm successful galectin-3 expression.

To induce CRISPR/Cas9-mediated deletion of Gal3, the LentiCRISPR v2 plasmid containing Cas9 and including a puromycin selection marker was purchased from Addgene (Plasmid #52961). A 20 nucleotide sgRNA sequence targeting exon 3 of mouse *lgals3* was designed using the website http://crispr.mit.edu/ (accessed 10 December 2015). The complementary sgRNA oligonucleotides (forward): 5′-caccgTCAAGGATATCCGGGTGCAT-3′ and (reverse): 5′-AAACATGCACCCGGATATCCTTGAC-3′, (synthesized by Integrated DNA Technology) were annealed and cloned into the LentiCRISPR v2 plasmid, and DNA sequences were verified. Clones were first verified by restriction digestion with BsmBI.

CRISPR-induced mutations were detected using a Surveyor Mutation Detection Kit (#706025, Integrated DNA Technologies) according to the manufacturer’s instructions. The frequency and type of mutations were determined by cloning the OP9 genomic DNA fragment spanning 600 bp around the gRNA DNA break site into pUC19 plasmid (#50005, AddGene) and sequencing 50 single colonies (primers: [forward]: 5′-AGGCCAGAACAAGACATGATACA-3′ and [reverse]: 5′-ACCAATGTCCCCTCCACTTG-3′). Serial dilutions in a 96-well plate were used to isolate and identify single colonies with bi-allelic homozygous mutations. Western blotting was done to detect complete Gal3 expression knockout. CRISPR/Cas9 mediated *lgals3* homozygous disruption occurred at frequencies of around 32% and resulted in three genotypes (Appendix A).

### 2.8. Western Immunoblot

For Western blotting, BCP-ALL cells were harvested according to the following method. Human BCP-ALL cells in co-culture with OP9 stromal cells have a dynamic interaction with the stroma: most of the different BCP-ALL samples migrate towards the stromal cells and adhere to the top. Many BCP-ALLs also migrate underneath the stroma to form cobblestone-forming areas. The OP9 stromal cells were firmly attached and spread out on gelatin-coated plates. The BCP-ALL cells proliferate underneath the stromal layer and then migrate back into the medium. Therefore, in such co-cultures, BCP-ALL cells from the overstanding medium and loosely attached to the top of the OP9 stromal cells can be harvested. Whole cell lysates were prepared by suspending cell pellets (on ice) for 20 min in RIPA buffer (1% NP-40, 0.1% SDS, 150 mM NaCl, 50 mM Tris, pH 8.0) supplemented with 1× complete EDTA-free protease inhibitor and 1× PhosStop (Roche, Basel, Switzerland). Lysates were separated by 4–20% SDS-PAGE and transferred to PVDF membranes (GE Healthcare, Piscataway, NJ, USA). Membranes were blocked with 5% BSA in 1× TBS with 0.1% Tween 20 for 1 h at room temperature with shaking. Primary antibodies for actin (sc-47778, Santa Cruz Biotechnology, Santa Cruz CA, USA), Galectin-1 (GTX101566) (GeneTex, Irvine, CA, USA), and galectin-3 (#125402, Biolegend, San Diego, CA, USA), were diluted 1:1000 or 1:2000 in 5% BSA and incubated for one hour at room temperature. Membranes were washed using 1× TBS-T and incubated with HRP-conjugated secondary antibodies diluted in 1× TBS-T with 5% BSA, washed, and developed using SuperSignal West Dura Chemiluminescent Substrate (#32106, Thermo Fisher Scientific).

### 2.9. Statistical Analysis

Biological experiments were analyzed by ANOVA using GraphPad Prism and Excel software. The value of *p* < 0.05 was considered to be statistically significant. Treatments were carried out in triplicate, and at least in two independent experiments with different BCP-ALL cells. Statistical tests used to analyze significance are indicated in each figure legend.

## 3. Results

### 3.1. Bone Marrow Stromal Cells as Source of Galectin-3

Pediatric patients with BCP-ALL are typically treated with a one-month regimen of induction chemotherapy, of which vincristine is usually one component. Thereafter, patient BM is examined for the presence of minimal residual disease (MRD). The absence of MRD at this time point is highly correlated with event-free and overall survival in both pediatric and adult BCP-ALL [22]. As shown in Appendix A, the reduction in CD19 and CD10 mRNA expression as hallmarks of BCP-ALL cells in these BM specimen [23] reflects the massive eradication of the leukemia cells. Interestingly, there is a marked increase in galectin-3 mRNA in the same samples, suggesting that chemotherapeutic treatment of the BM, including leukemia cells and cells in the leukemia microenvironment, induces high amounts of galectin-3. Thus, at the end of induction therapy, the BM is likely to contain high levels of galectin-3. Higher levels of galectin-3 (*LGALS3*) mRNA also correlate with a higher probability of being MRD-positive at the end of such induction therapy (Appendix A).

Our previous studies showed elevated levels of galectin-3 in BCP-ALL patient bone marrow plasma compared to normal controls [11]. The bone marrow tumor niche where BCP-ALL develops consists of different stromal cell types, some of which contribute in a major way to the survival and growth of the leukemia. The mesenchymal stromal cells (MSC) are such key components [24,25]. To determine if these cells make galectin-3, we grew MSC out from primary human bone marrows. FACS analysis of CD45^neg^, SSEA4^pos^, CD271^pos^ cells confirmed that this entire population was positive for Gal3 expression (Figure 1A, MSC-1 and MSC-2). This result is consistent with those reported for bone marrow stromal cells from patients with acute myeloid leukemia [26,27]. We also compared the ability of such MSC to support BCP-ALL cells next to a human MSC cell line and a murine bone marrow MSC cell line OP9. Based on viability, all three MSC were able to keep the BCP-ALL cells alive in such co-culture systems (Figure 1B, Figure 1C, right panel). However, interestingly, the OP9 cells allowed more cell proliferation compared to the MSC and were also superior in providing protection against chemotherapy treatment with vincristine (Figure 1C, left panel). Because OP9 cells allow better BCP-ALL growth, have a longer life span than primary MSC and are more consistent in their own growth, we used these cells for subsequent experiments.

### 3.2. Gal3 Made by Stroma Is a Major Source for BCP-ALL Cells

We also examined five different primary BCP-ALL samples by FACS, finding that a variable percentage of cells in the total population were positive for Gal3. One sample was able to grow in co-culture with OP9 cells (Appendix A). To confirm that Gal3 can be taken up in such co-cultures we kept US7 and TXL2, two PDX-derived BCP-ALLs, for two days without stroma. Based on Western blots, these cells lacked Gal3 but acquired a Gal3 signal when co-cultured with OP9 wild type stromal cells (Appendix A). Moreover, we previously showed that OP9 stromal cells secrete galectin-3 into the medium [11]. To examine the effect of stromal-produced galectin-3, we used CRISPR/Cas9 to knock out Gal3 in those cells (strategy in Appendix A). We used single-cell cloning to isolate clones with bi-allelic Gal3 knockout (Appendix A). Compared to empty vector-transduced OP9 (OP9-EV), the OP9-Gal3-KO cells had a minimal signal using FACS (Figure 2A). We then used the two different OP9 genotypes (KO and EV) for co-culture with two different human BCP-ALLs, US7 and TXL2. As shown in Appendix A, when these leukemias were harvested from above the OP9-Gal3-KO cells, no Gal3 was detected by Western blot. In contrast, when the same cells were co-cultured with OP9-EV cells, Gal3 protein was detected, indicating this was murine galectin-3. Although antibodies used to detect Gal3 are not species-specific, human and mouse galectin-3 can be distinguished because they differ in molecular weight, as shown in Appendix A. These results are consistent with our previous report, in which we found little to no *LGALS3* mRNA in these cells under steady-state growth conditions and stromal Gal3 uptake by BCP-ALL cells [12].

### 3.3. Effect of Gal3 Loss on BCP-ALL Homeostasis

We first examined the effects of loss of stromal-produced Gal3 on BCP-ALL cell viability and proliferation. Both US7 and TXL2 grew normally in vitro when co-cultured with both Gal3-KO and control OP9 cells over a two-week co-culture: viability (Appendix A) and proliferation (Appendix A) in the absence of stress were normal and, accordingly, no change was detected in the BCP-ALL cell cycle as determined by FACS-assisted measurement of DNA content (Appendix A). These observations indicate that, under steady-state conditions, stromal Gal3 expression does not regulate BCP-ALL cell survival and proliferation.

BCP-ALL cells depend on supportive stromal cells for sustenance and expansion in culture in part because they migrate towards specific chemokines secreted by these cells and then adhere to and migrate underneath the stroma. Because Gal3 has been implicated in regulating both normal and tumor cell motility, we examined if extracellular Gal3 levels can also modulate BCP-ALL movement. To measure the association of the leukemia cells to Gal3-deficient stroma, we determined the number of living floating and adhered BCP-ALL cells via Trypan blue exclusion. This assay showed that significantly lower numbers of BCP-ALL cells were found to be adhered to OP9-Gal3-KO cells than to control OP9 stroma after a 24 or 72 h co-culture and, conversely, that more cells remained floating in the medium (Figure 3A, US7; Figure 3B, TXL2). To demonstrate that this was Gal3-dependent, we also reconstituted the OP9-Gal3-KO cells with human Gal3. Although expression levels were lower than that of WT OP9 (Figure 2A) it was sufficient to restore association with the OP9 cells (Figure 3C). We also performed a Transwell migration assay and found that fewer US7 cells migrated down to OP9 cells with no Gal3 production than to control OP9 cells (Figure 3D). These findings point to a key role for stromal-produced Gal3 in mediating the migration and adhesion of BCP-ALL cells to the crucial protective stromal components of the tumor microenvironment.

### 3.4. Depletion of Gal3 Increases BCP-ALL Cells Response to Drug Treatment

We next examined if stromal-produced Gal3 regulates drug sensitivity of BCP-ALL cells. BCP-ALL cells were cultured with either OP9-Gal3-KO or OP9-EV control stroma in the presence of 5 nM vincristine, part of the standard cytotoxic therapy in BCP-ALL (Vin, to treat Ph- US7 cells) or 20 nM of the targeted tyrosine kinase inhibitor nilotinib (Nilo, to treat Ph+ TXL2 cells) for >15 days. Cell viability was measured every three days during drug treatment and is presented as percent of total. Interestingly, as shown in Figure 4A,B, deficiency of Gal3 in BCP-ALL cells increased the sensitivity of the BCP-ALL cells to the chemotherapy when compared to OP9-EV control. In both treatment groups, following a sharp decline, cell viability recovered around day five (Figure 4A,B). However, the viability of leukemia cells after day five was significantly lower in BCP-ALL cells expanded with Gal3-deficient stroma when compared to control OP9, suggesting that extracellular Gal3 can provide considerable chemoprotection for BCP-ALL cells. As expected, the viability of BCP-ALL cells in the absence of chemotherapy (control DMSO treated co-cultures) remained unaffected regardless of the Gal3 status of the stroma.

Leukemia cell proliferation (Figure 4C,D) was inhibited by vincristine after a few days of drug treatment. BCP-ALL cells co-cultured with Gal3-deficient stroma were more affected by day six, suggesting that BCP-ALL cells lacking stromal-produced Gal3 responded better to chemotherapy. The expression of human Gal3 in the OP9 Gal-3 deficient cells was able to partly rescue both viability and proliferation (Figure 4E,F and Figure 4G,H, respectively). We conclude that Gal3 contributes to the microenvironment-mediated support against conventional drug treatment in BCP-ALL cells, and that inhibiting Gal3 may enhance BCP-ALL cell sensitivity to standard chemotherapy agents.

### 3.5. Novel Galectin-3 Inhibitors-Effect on BCP-ALL Cell Function

The design of specific small molecule inhibitors of Gal3 has been challenging because of the high amino acid sequence homology between members of the galectin family as well as the weak binding of carbohydrates and lectins, and low bioavailability [28]. We recently reported the development of a series of taloside-based compounds [21], Cpd14-Cpd18, with significant specificity for Gal3. The lead compound Cpd14 was able to significantly reduce viability in US7 and TXL2 in a 24 h assay at 250 μM. We next tested the effects of Cpd14 and Cpd17 on important Gal3-regulated activities. BCP-ALL cells adhere to fibronectin (FN) in an integrin-dependent manner [28,29]. As shown in Figure 5A,B, both US7 and TXL2 cells can adhere to fibronectin-coated plates (controls). We compared the effect of Cpd14 on FN-mediated adhesion of the BCP-ALLs to that of β-lactose, which is a natural competitor of Gal3 in binding to carbohydrate structures. Although Cpd14 is a monosaccharide-based compound, it was able to inhibit cell adhesion of TXL2 at an approximately 12-fold lower concentration than the disaccharide lactose (Figure 5B). Interestingly, Cpd14 had a more potent inhibitory effect on migration of both US7 and TXL2 cells. Migration to both OP9 cells as well as to the chemokine SDF1-α secreted by stromal cells (Figure 5C,D) was inhibited. We also tested Cpd17, of which the IC_50_ was significantly lower than Cpd14 (83 μM and 307 μM) [30]. As shown in Figure 5E, in agreement with this, much lower amounts were needed to inhibit adhesion of BCP-ALL cells to OP9 stroma.

Because both Cpd14 and Cpd17 were able to inhibit BCP-ALL cell proliferation, we also tested their effects on the cell cycle. As indicated by Figure 6A,B, the most clear effect of Cpd14 was to reduce the number of cells in S-phase (Figure 6A, right panel, arrow). Compared to DMSO-treated cells, the percentage of cells in S phase were reduced following Cpd17 treatment, but this was not significant. Compared to the pan-Aurora kinase inhibitor PHA-739358, which also reduces the % of cells in S-phase but causes accumulation in G2, the effect was small (Figure 6C). To investigate if BCP-ALL cells lose viability because of apoptosis, we measured AV/PI expression using FACS. Compared to DMSO-treated controls, a 48 h treatment with 250 or 500 μM Cpd14 caused an increase in AnnexinV/PI^pos^ US7 or TXL2 cells [not shown]. This also applied to Cpd17 but at lower concentrations; a 24 h treatment with 20 μM Cpd17 increased the number of AnnexinV/PI^pos^ US7 cells (Figure 6D).

### 3.6. Chemotherapeutic Treatment of BCP-ALL Combined with Cpd17

We next determined the IC_50_ for Cpd17 in five different human ALLs including TXL2 (diagnosis, Ph-positive), US7 (diagnosis, no karyotype abnormalities), ICN06 (diagnosis, ETV6-Runx1), ICN3 (relapsed, MLL-AF4, infant ALL) and ICN13 (diagnosis, MLL-AF4, infant ALL). As shown in Figure 7A, different BCP-ALLs had similar IC_50_ values for Cpd17. When tested over an eight-day period in the presence of OP9 cells (Figure 7B–E, top, cell counts; bottom, viability), Cpd17 used at the approximate IC_50_ value suppressed BCP-ALL cell growth by 50 % on day three, which is consistent with results obtained with Alamar Blue in Figure 7A. Cpd17 continued to suppress BCP-ALL cell growth after day three (Figure 7B–E top) and thus, clearly, the compound is cytostatic at this concentration. In addition, viability of the cells decreased starting at day three (Figure 7B–E bottom panels), indicating that the compound is cytotoxic at this concentration.

Next, we tested the possibility of a combination treatment with Cpd17 on a number of different BCP-ALL cells, measuring both cell counts (Figure 8) and cell viability (Appendix A) over a 23-day treatment of leukemia cells in co-culture with OP9 stroma to allow detection of the emergence of relapse. We first determined the lowest effective concentration of Cpd17 for a drug combination treatment of US7 cells. US7 cells were treated with 2 nM vincristine alone, and with either 1 or 5 µM Cpd17 alone, or a combination of vincristine and Cpd17, as indicated in the figure. As expected, (Figure 8A,B), Cpd17 at concentrations of 1 μM and 5 μM did not affect US7 cell growth compared to control (DMSO). Vincristine at 2 nM suppressed proliferation of the cells, but after about day 10, relapses emerged in that the cells resumed growth in spite of the continued presence of vincristine. Interestingly, in the combination treatment of 2 nM vincristine with 1 μM Cpd17, relapse was clearly suppressed, although not eliminated (Figure 8A). In contrast, when 5 μM Cpd17 was combined with 2 nM vincristine, cells were no longer able to divide, even after an extended period of observation (Figure 8B).

Similar to US7, the growth of ICN13 was not affected by 1 or 5 μM Cpd17 (Figure 8C,D). Cell division of ICN13 was inhibited less than US7 by treatment with 2 nM vincristine. However, also with this BCP-ALL, the combination of Cpd17 with vincristine showed synergistic cytotoxic effects in ICN13 cells, but relapse did emerge with the drug combination. Viability also dropped in the first 9–10 days, but cells become resistant.

ICN3 and ICN06 BCP-ALLs were not very responsive to 2 nM vincristine (Figure 8E,F, compare DMSO to 2 nM vincristine) and Cpd17 alone at 5 μM was not cytostatic either. Nonetheless, when 2 nM vincristine was combined with 5 µM Cpd17, growth was suppressed over at least 10 days of treatment. The viability of the cells also decreased in the first 9–10 days of combination treatment, after which resistance developed (Appendix A).

TXL2 BCP-ALL is driven by an oncogenic Bcr/Abl fusion protein and is treated with the targeted tyrosine kinase inhibitor nilotinib. Nilotinib on its own was not cytotoxic, but did inhibit the proliferation of these leukemia cells (Figure 8G, compare DMSO and nilotinib). The addition of 40 µM Cpd17 to the nilotinib further reduced proliferation and modestly reduced viability of the cells (Figure 8G and Appendix A).

## 4. Discussion

Gal3 has been implicated in numerous pathologies [15,31], is present at different subcellular locations [32,33,34,35,36,37] and has almost 250 cell-surface binding partners [38]. Because of this, the exact mechanism by which stromal produced extracellular Gal3 protects BCP-ALL cells against the conventional chemotherapy drugs vincristine and nilotinib, as found in our study, is difficult to determine. However, survival and proliferation of BCP-ALL cells is strongly dependent on their ability to migrate and adhere to specifically protective microenvironmental niche cells in the bone marrow [39]. When such interactions are inhibited, for example by using AMD3100, next generation CXCR4 inhibitors or integrin-blocking strategies, BCP-ALL cells are mobilized and become easier to eradicate with conventional chemotherapy [5,7,28,40,41,42]. We here show that extracellular Gal3 produced by stromal cells functions as one of the signals to BCP-ALL cells which regulates leukemic cell adhesion and migration. Therefore, we argue that one of the mechanisms through which reduction in extracellular Gal3 activity makes BCP-ALL cells more vulnerable to drug treatment is through an indirect route, by interfering with so-called cell adhesion-mediated drug resistance [43,44].

How Gal3 produced by stromal cells can promote BCP-ALL migration is currently not clear, but it could cluster key glycoprotein binding partners such as integrins on the plasma membrane [45] and promote cell polarization and subsequent directed migration. A second possibility is that Gal3 from stroma regulates intracellular events after its uptake by the BCP-ALL cells, as we have previously shown [12] and confirmed (Appendix A). For example, in prostate cancer cells, Gal3, through stabilizing focal adhesion kinase at focal adhesions, can promote cancer cell motility [46]. In HeLa cells, Gal3-mediated activation of the Erk1/2 pathway was shown to be necessary for cell migration [13]. One intracellular mechanism through which Gal3 could polarize cells is through its interaction with Myh2a, an interaction previously reported by Nakajima et al. [47]. Using a recombinant Gal3 affinity column and confirmed through co-immunoprecipitation, we have also detected a direct protein-protein interaction between NMIIA (MYH9) and Gal3. Because NMIIA inhibition using blebbistatin resulted in a significant reduction in BCP-ALL adhesion to fibronectin, and migration toward SDF1α as well as OP9 cells, the intercellular interaction between these proteins could regulate these BCP-ALL activities (results not shown).

Pharmacologic targeting of Gal3 has been proposed as an attractive novel therapy for acute myeloid leukemia [48,49,50,51]. However, the specific targeting of the Gal3 protein using drugs or drug-like compounds has been a considerable challenge due to the highly conserved carbohydrate recognition domain shared by all galectins. In addition, galectins have a weak binding affinity for their carbohydrate ligands. Cpd14 and Cpd17 were developed as novel, talopyranoside-based [21], small molecule antagonists, and from the perspective of such challenges, Cpd17 as a monosaccharide-based compound did have an impressive activity on BCP-ALL cells. Apart from the clear effect on cell migration, the compounds also had cytostatic and even cytotoxic effects when used alone.

Our knock out experiments of Gal3 in the stromal cells provided proof-of-principle that a reduction in Gal3 (activity) with its origin in the stromal cells would be beneficial, for example in reducing the chance of persistence of MRD in the induction chemotherapy phase of treatment of BCP-ALL. However, Gal3 is a relatively abundant protein, and ng/mL amounts have been measured in BCP-ALL bone marrow plasma [12], as well as in the serum and plasma of patients with carcinomas (for example, [52]). We note that the knockout of Gal3 in stroma, as studied here, permanently eliminates production of stromal galectin-3, whereas pharmacological galectin-3 inhibition is temporary. Because some of our pharmacological inhibition assays were done in the absence of OP9 cells, all pharmacologically inhibitable Gal3 would have to be that which is endogenous to BCP-ALL cells. Those levels of Gal3 are low in non-stressed BCP-ALL cells, but such cells do synthesize Gal3 de novo when subjected to stress [11,12]. Taken together, effective galectin-3 inhibitors would probably need to act both intracellularly to inhibit cell-intrinsic galectin-3 as well as inactivate the abundant galectin-3 produced by stromal cells in the tumor microenvironment.

Finally, for Cpd14 and Cpd17, we cannot exclude the possibility of significant off-target effects on cell viability, in particular because high amounts of compounds were needed. On the other hand, the abundance of galectin-3 may require equally high amounts of inhibitor if a 1:1 stoichiometry is needed for a biological effect. We conclude that these promising compounds will require further optimization before they are suitable for in vivo preclinical studies. For example, we found that Cpd17 is not chemically stable, and even in 100% DMSO stored at −20 °C, its activity markedly diminished. We also were unable to test Cpd17 in animal experiments because it is very hydrophobic and it was not possible to find a biocompatible solvent. Nonetheless, by a careful titration of the drugs used in the chemotherapy combination experiments, we were able to show useful effects of Cpd17 at low micromolar (5 μM) amounts when combined with doses of vincristine that were unable to prevent BCP-ALL cell proliferation. This warrants further development of Cpd17 or derivatives to increase solubility. Since Gal3 is implicated in other malignancies, the impact of understanding how it regulates motility and invasion will not be limited to BCP-ALL. Importantly, our studies show that it is feasible to target the crosstalk between the tumor microenvironment and BCP-ALL cells via carbohydrate-lectin interactions, and show that Gal3 inhibition could be a novel approach to treat BCP-ALL.

## Figures and Tables

**Figure 1 ijms-22-12167-f001:**
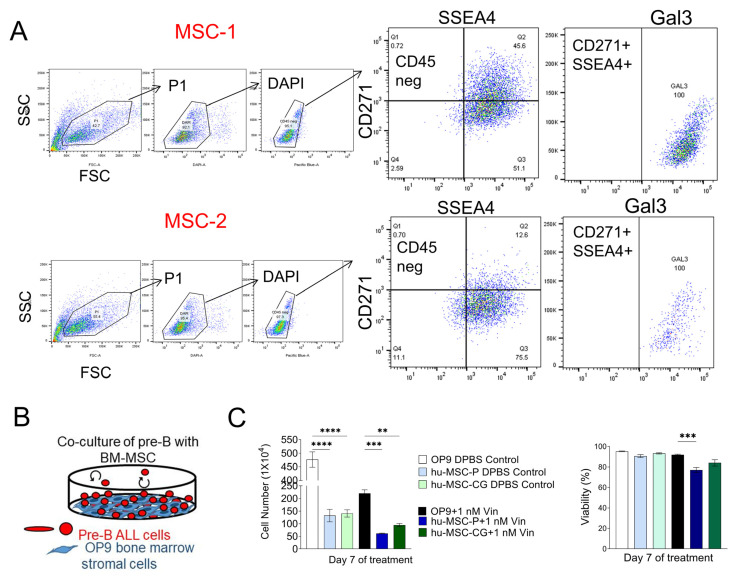
Bone marrow stromal cells produce galectin-3. (**A**) MSC from two different normal BMT donor screens MSC-1 and MSC-2 were grown out as adherent cells. FACS analysis on the indicated populations using the gating strategy shown in the figure. (**B**) Schematic co-culture system. Stromal cells have been mitotically inactivated. They support the leukemia cells but no longer can divide. (**C**) Comparison of human primary bone marrow MSC with mouse bone marrow MSC OP9 cells for ability to protect human BCP-ALL US7 cells against chemotherapy. Right panel: viability (viable cell number/total cell number × 100) determined using Trypan blue. Left panel: total cell number. Stromal cells were mitotically inactivated by treatment with 10 μg/mL mitomycin C. Analysis on day seven of 1 nM vincristine treatment (comparison with OP9 cells, triplicate samples harvested in a single experiment. One-way ANOVA, Tukey’s multiple comparisons). hMSC-1 and hMSC-CG, primary and immortalized human MSC. ** *p* < 0.01, *** *p* < 0.001, **** *p* < 0.0001.

**Figure 2 ijms-22-12167-f002:**
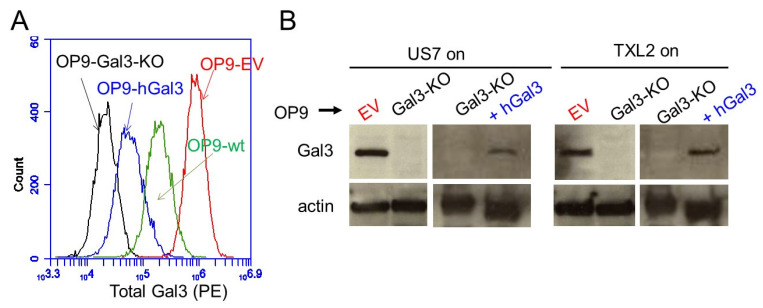
Protective stromal cells provide Gal3 to BCP-ALL cells. (**A**) Total Gal3 expression in 10,000 fixed and permeabilized OP9 stromal cells of the indicated genotypes. (**B**) Western blot of the indicated BCP-ALLs grown for >four days with OP9-EV, Gal3-KO or Gal3-KO + hGal3 cells. BCP-ALL cells were harvested from the medium.

**Figure 3 ijms-22-12167-f003:**
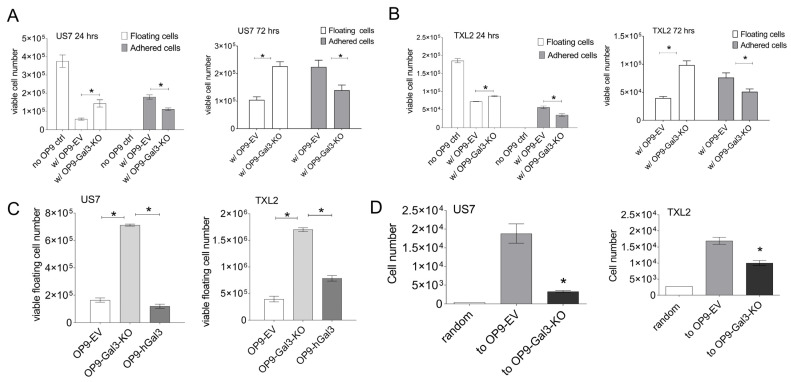
Gal3 deficiency in stromal cells impairs BCP-ALL cell adhesion and migration. Adhesion of BCP-ALL cells to Gal3-deficient OP9 and control stroma presented as number of viable cells. (**A**) US7 cells after 24 and 72 h of migration and adhesion. (**B**) TXL2 cells at 24 and 72 h. (**A**,**B**) BCP-ALL cells in the supernatant (floating) and attached to OP9 stroma (adhered) were both determined. (**C**) Adhesion of US7 or TXL2 cells to the OP9 cells of the indicated genotype measured after 24 h. OP9-hGal3: OP9 Gal3-KO cells expressing hGal3. (**D**) Migration of US7 or TXL2 cells over a 4 h period toward OP9 Gal3-KO stroma and control OP9-EV cells measured using a Transwell assay. Motility in the presence of medium was used as readout for random (spontaneous) migration. Error bars, mean ± SEM of duplicate values in (**C**,**D**) or triplicates for (**A**,**B**) samples. (**A**,**B**) two-way ANOVA, Šidák’s multiple comparison test; (**C**,**D**) one-way ANOVA, Tukey’s multiple comparison test. Adjusted *p*-values. * *p* < 0.05.

**Figure 4 ijms-22-12167-f004:**
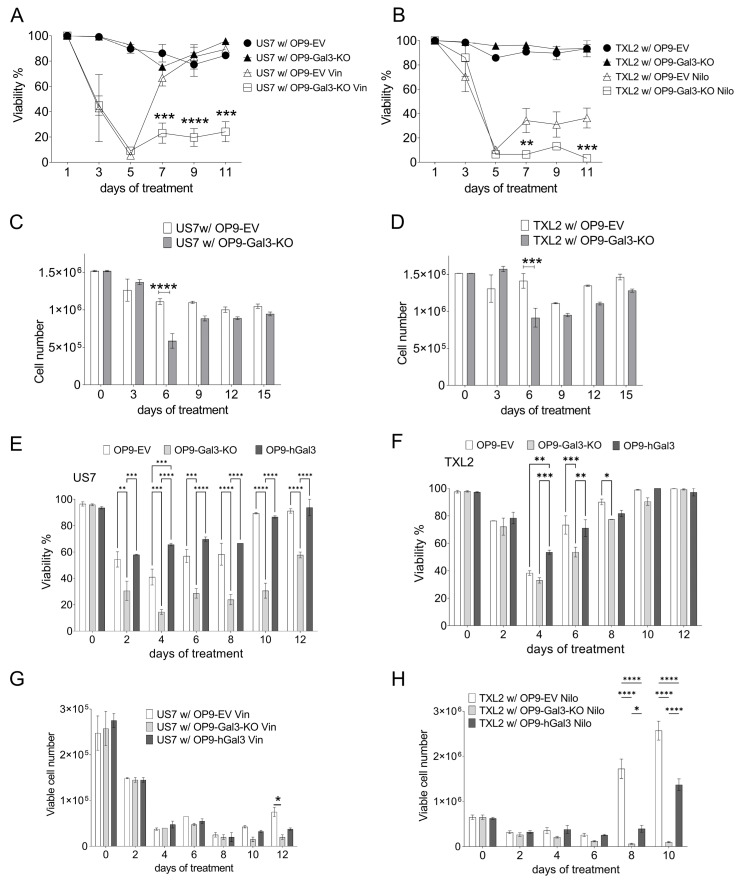
Depletion of Gal3 in stromal cells significantly enhances BCP-ALL cell response to chemotherapy. (**A**) Viability of US7 cells grown on the indicated OP9 cells treated with solvent as control, or with 2 nM vincristine. (**B**) Viability of TXL2 cells plated on different OP9 cells and treated with 20 nM nilotinib. Panels (**A**,**B**): 2-way ANOVA comparing samples per time point, Tukey’s multiple comparison test; only statistical analysis of comparisons between vincristine-treated cells on OP9-Gal3-KO and OP9-EV are shown. (**C**) Cell numbers of analysis similar to panel (**A**). (**D**) Cell numbers of analysis similar to panel **B**. Panels (**C**,**D**): 2-way ANOVA comparing samples at each time point, Šidák’s multiple comparison test. (**E**) Viability of US7 cells grown on the indicated OP9 cells treated with solvent as control, or with 2 nM vincristine (**F**) Viability of TXL2 cells plated on the different OP9 cells and treated with 20 nM nilotinib. (**G**) Cell numbers of samples in panel (**E**). (**H**) Cell numbers of samples in panel (**F**). OP9-Gal3 KO cells reconstituted with human Gal3 are indicated as hGal3. Panels (**E**–**H**): 2-way ANOVA comparing samples at each time point, Tukey’s multiple comparison test. Only values with significant differences are shown. Other comparisons, not significantly different. Panels (**A**–**D**): error bars, mean ± SEM of 3–4 replicates per time point. Panels (**E**–**H**): two replicates per time point. * *p* < 0.05, ** *p* < 0.01, *** *p* < 0.001, **** *p* < 0.0001. Viability is the percentage of Trypan blue excluding cells/total cells.

**Figure 5 ijms-22-12167-f005:**
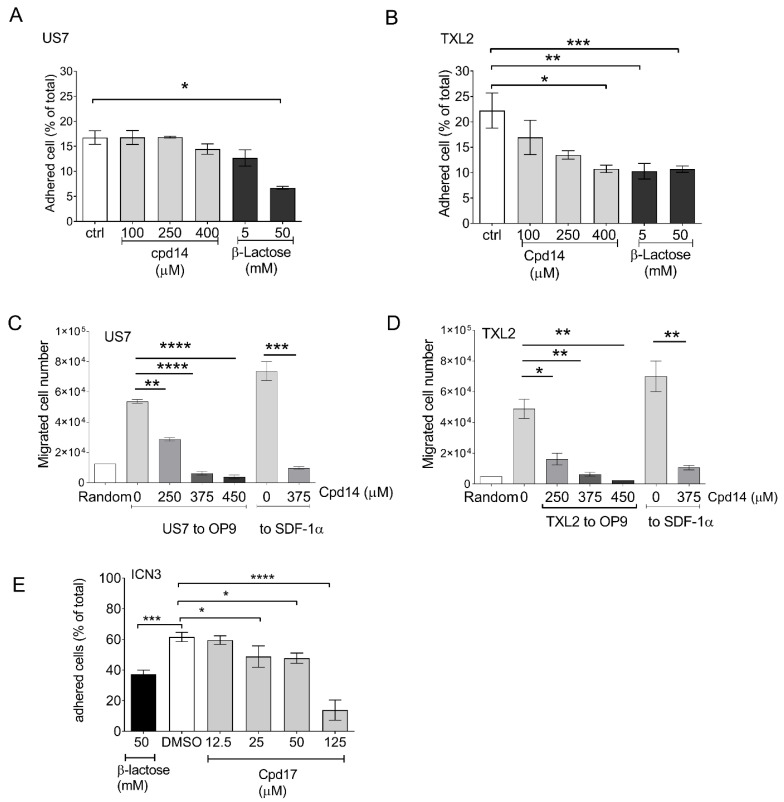
Effects of Cpd14 or 17 on BCP-ALL migration and adhesion. Adhesion of US7 (**A**) or TXL2 cells (**B**) to fibronectin-coated wells with and without the indicated amounts of Cpd14 in a 30-min assay. Migration of US7 (**C**) or TXL2 (**D**) to OP9 or to 200 ng/mL of the chemokine SDF-1α (24 and 4 h, respectively) in a Transwell assay when treated with the indicated concentrations of Cpd14. (**E**) Adhesion of ICN13 cells to OP9 stroma measured after 24 h in the presence of the indicated concentrations of Cpd17. Cells in suspension and adherent to (above and underneath) the OP9 cells were harvested and counted by Trypan blue exclusion. Results are presented as percentage of living [Trypan-blue excluding] leukemia cells adhering to OP9. Error bars, mean ± SEM of triplicate values for panels (**A**,**B**,**E**) or duplicate wells for panels (**C**,**D**); ctrl, DMSO control. One-way ANOVA, Dunnett;’s multiple comparison test. Adjusted *p*-values. * *p* < 0.05, ** *p* < 0.01, *** *p* < 0.001, **** *p* < 0.0001.

**Figure 6 ijms-22-12167-f006:**
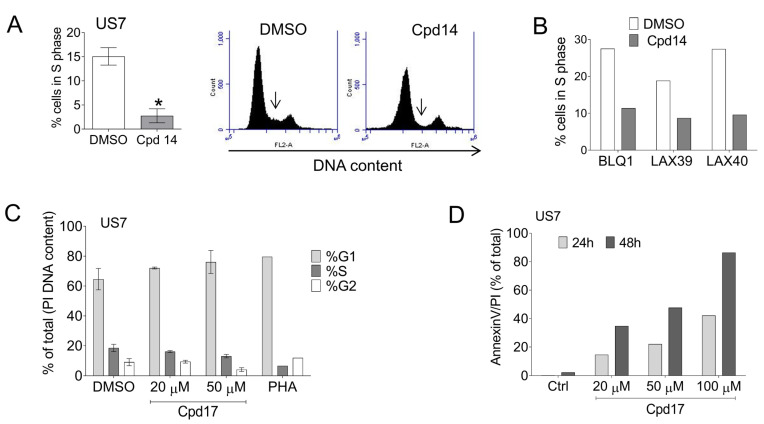
Analysis of effects of Cpd14 and 17 on cell cycle and apoptosis. (**A**) Cell cycle of US7 cells measured by DNA PI staining using FACS, following 48 h exposure to 250 µM Cpd14. Left: percentage of BCP-ALL cells in S phase; Right: representative image, arrow points to S phase DNA content. Error bars, mean ± SEM of duplicate measurements. * *p* < 0.05, unpaired *t*-test, two-tailed. (**B**) Percentage of BCP-ALL cells as indicated in S phase after 48-h treatment with 250 μM Cpd14. Single samples, 3 different BCP-ALLs. (**C**) Cell cycle analysis on US7 cells treated with Cpd17 using FACS and PI DNA staining. Duplicate measurements, 2-way ANOVA, differences not significant. PHA, 1 μM PHA-739358 (danusertib), single sample. (**D**) Percentage of apoptotic US7 cells based on AnnexinV/PI positivity after treatment with Cpd17.

**Figure 7 ijms-22-12167-f007:**
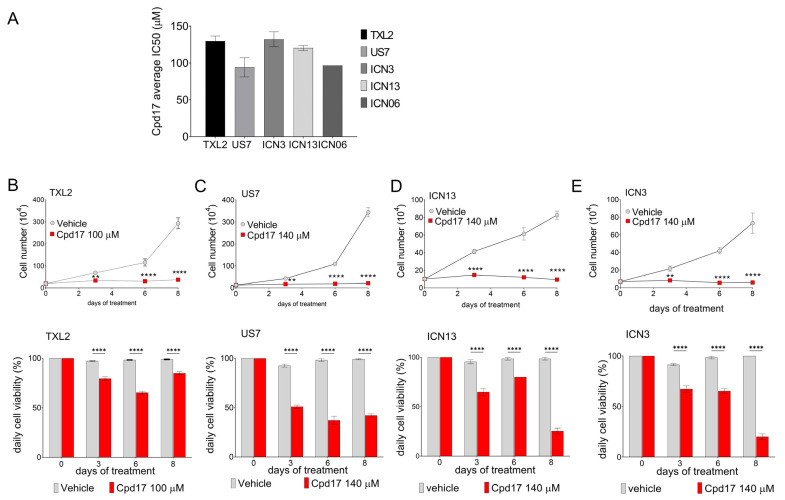
Determination of IC_50_ values for Cpd17. (**A**) For IC_50_ determination, cells were incubated with different concentrations of Cpd17 for three days without OP9 cells. Cell counts were done using Alamar blue. Error bars, mean ± SEM of independent triplicate IC_50_ determinations. (**B**–**E**) Different ALLs as indicated were co-cultured with OP9 cells for eight days and treated with ≈IC_50_ concentrations of Cpd17. Top panels, cell numbers; lower panels viability determined by Trypan blue exclusion. Representative results from three independent experiments. Note: cell viability is defined here as the number of Trypan blue excluding cells in Cpd17 treatment group/number of Trypan blue excluding cells in group treated with solvent DMSO at each time point. Error bars, mean ± SEM of triplicate samples. Two-way ANOVA, Šidák’s multiple comparison test. Each point compares vehicle to Cpd17 on the same day. ** *p* < 0.01, **** *p* < 0.0001.

**Figure 8 ijms-22-12167-f008:**
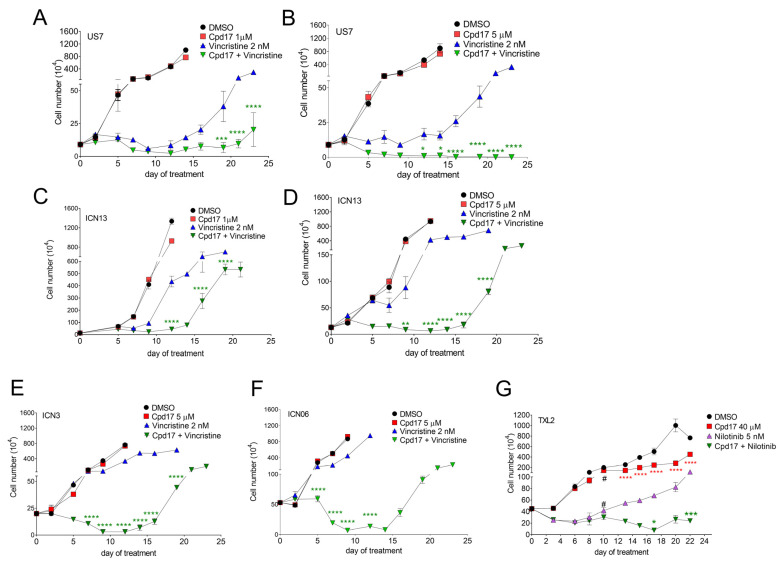
Proliferation of BCP-ALLs under combination treatment with Cpd17. Cells were treated with DMSO, mono-treatment or combination as indicated. Fresh drugs were added with each change of medium. Cell counts by Trypan blue exclusion in DMSO and Cpd17 groups were terminated (on day 12–14 for DMSO and Cpd17-only samples, later in vincristine-only resistance) when growth exceeded well capacity in panels (**A**–**F**). Results presented are average of three replicates +/− SD of one experiment. Note the discontinuity of the *Y*-axis. (**A**) US7 treated with 1 μM Cpd17 and 2 nM vincristine. Vincristine alone or vincristine + Cpd17 compared to DMSO, significantly different starting at d7 (not indicated in the figure). (**B**) US7 treated with 5 μM Cpd17 and 2 nM vincristine. Vincristine alone or vincristine + Cpd17 compared to DMSO, significantly different starting at d9. (**C**) ICN13 treated with 1 μM Cpd17 and 2 nM vincristine. Vincristine alone or vincristine + Cpd17 compared to DMSO, significantly different starting at d7. Vincristine compared to vincristine + Cpd17 significantly different on d9 and d12 as indicated in the figure [green asterisk]. (**D**) ICN13 treated with 5 μM Cpd17 and 2 nM vincristine. Vincristine alone or vincristine + Cpd17 compared to DMSO, significantly different starting at d5. Also significant differences between vincristine alone and vincristine + Cpd17 on d5, d7, d9 and d12 as indicated. (**E**) ICN3 treated with 5 μM Cpd17 and 2 nM vincristine. Difference between DMSO and vincristine alone or vincristine + Cpd17 significant starting from d9. Vincristine alone and vincristine + Cpd17, significantly different on d5, d7, d9, d12 [green asterisk]. (**F**) ICN06 treated with 5 μM Cpd17 and 2 nM vincristine. Vincristine alone and vincristine + Cpd17, significantly different on d5, d7, d9 [green asterisk]. (**G**) TXL2 treated with 40 μM Cpd17 and 5 nM nilotinib. Cpd17 was administered at 10 μM on day 0, increased to 20 μM on day six and further increased to 40 μM on day 10 as indicated by #, because drug combination effects were minimal at 10 or 20 μM. Nilotinib alone and nilotinib + Cpd17, significantly different from DMSO starting at d6. Nilotinib compared to nilotinib + Cpd17 significantly different on d17, d22 [green asterisks]. Cpd17 compared to DMSO different from d13–d23 [red asterisks]. Panel G: 2-way ANOVA, Tukey’s multiple comparison test. Panels (**A**–**F**): 2-way ANOVA, Šidák’s multiple comparison test for vincristine versus vincristine + Cpd17; adjusted *p*-values. * *p* < 0.05, ** *p* < 0.01, *** *p* < 0.001, **** *p* < 0.0001.

## Data Availability

Data analyzed in Appendix A can he acceesed here https://www.ncbi.nlm.nih.gov/geo/query/acc.cgi?acc=GSE67684, https://www.ncbi.nlm.nih.gov/geo/query/acc.cgi?acc=GSE11877, https://www.ncbi.nlm.nih.gov/geo/query/acc.cgi?acc=GSE28460.

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
