# Peer review of "Overcoming Microenvironment-Mediated Chemoprotection through Stromal Galectin-3 Inhibition in Acute Lymphoblastic Leukemia"

_ijms, 2021, doi:10.3390/ijms222212167_

Round 1

Reviewer 1 Report

Somayeh et al. evaluated the effect of gelectuin-3 on the proliferation and chemoprotection of ALL cells cultured on the MSC cells. The concept of this report is very interesting and important for overcoming ALL, however, I am unable to evaluate these data because of the lack of appropriate statistical assessment and description.

Major points.

  1. The appropriate statistical assessment should be done and describe in every figure or in the method section for the evaluation of the result of the experiments.

For example, in figure 2A and B, the multiple independent experiments on the Gal3 expression of the target cells and the appropriate comparison of MFI should be included in the analysis to compare the expression level of the Gal3. In the figure 3, the statistical evaluation among the 3 conditions should be done by ANOVA test, not by repeating t-test. There are no explanation of the method of statistical analysis and justification of assessment with comparing the duplicate wells or single sample in figure 5C, 5D, 6C and 6D. In the Figure6B, the shape of the representative DNA content analysis seems to be inappropriate to quantify the cell cycle. Authors should explain the method of quantification. There is no assessment of statistical analysis in the figure 7 and 8 to conclude the hypothesis of the authors.

  1. In this paper, there is no clear evidence and explanation to insist that the difference of Gas3 were caused only by uptake by BCP-ALL cells from MSCs in the co-culture system if Gal3 protein synthesized endogenously in BCP-ALL cells. It would be more informative if the Gal3 from mouse oriented OP9 cells can be distinguished from Gal3 from human ALL cells using the difference of the specific origin, mouse and human.

And considering the expression of Gal3 in the ALL cells, it is unnatural no Gal3 was detected by Western blot from leukemic cells cultured above the OP9-Gal3-KO cells in the figure 2C. Authors should explain that including the method of separating OP9 cells and ALL cells after co-culture.

Author Response

Major points.

Comment 1: The appropriate statistical assessment should be done and describe in every figure or in the method section for the evaluation of the result of the experiments.

For example, in figure 2A and B, the multiple independent experiments on the Gal3 expression of the target cells and the appropriate comparison of MFI should be included in the analysis to compare the expression level of the Gal3. In the figure 3, the statistical evaluation among the 3 conditions should be done by ANOVA test, not by repeating t-test. There are no explanation of the method of statistical analysis and justification of assessment with comparing the duplicate wells or single sample in figure 5C, 5D, 6C and 6D. There is no assessment of statistical analysis in the figure 7 and 8 to conclude the hypothesis of the authors.

Reply: In our study we did not perform quantitation on the FACS results for the different Galectin-3 levels but we also did not make claims about the levels of Galectin-3 in the different OP9 cells that would require substantiation by such analysis. Regarding Figure 3 we agree with the reviewer and we apologize for the lack of statistical analysis for Figures 7 and 8 and the incomplete description of the statistical analysis of the other figures including 5C and 5D. 6C but not 6D was also done on duplicates. For cell cycle we analyzed duplicates statistically when the same experiment was performed on duplicates at the same time but not when experiments were done at different time points or with different BCP-ALLs. The original statistical analysis had been done by different postdoctoral fellows who performed the actual experiments. Dr. Yang, one of our co-authors and a computational biologist, has now re-analyzed all statistics. We have updated the figures and added information to the figure legends as to which statistical test was used.

Comment 2: In the Figure 6B, the shape of the representative DNA content analysis seems to be inappropriate to quantify the cell cycle. Authors should explain the method of quantification.  

Reply: The PI plot of TXL2 cells treated with DMSO seems irregular. Although the experiment was done with biological duplicates, unfortunately, we were not able to find a different matched plot done at the same time, as this experiment was performed several years ago. We therefore deleted this analysis from the figure and instead show cell cycle results from three different BCP-ALLs.

Comment 3: In this paper, there is no clear evidence and explanation to insist that the difference of Gas3 were caused only by uptake by BCP-ALL cells from MSCs in the co-culture system if Gal3 protein synthesized endogenously in BCP-ALL cells. And considering the expression of Gal3 in the ALL cells, it is unnatural no Gal3 was detected by Western blot from leukemic cells cultured above the OP9-Gal3-KO cells in the figure 2C. Authors should explain that including the method of separating OP9 cells and ALL cells after co-culture.

Reply: The reviewer raises an interesting question. We have included more Western blots in a new Figure S2 that are consistent and indicate that under non-stressed conditions, all Galectin-3 in the co-culture detectable by Western blot is mouse Galectin-3 [also see below- comment 4]. This is also consistent with previous experiments which we published. We have included this in the manuscript. We have also included additional information in the revised manuscript] to explain how we harvest BCP-ALL cells in the co-culture system. 

Comment 4: It would be more informative if the Gal3 from mouse oriented OP9 cells can be distinguished from Gal3 from human ALL cells using the difference of the specific origin, mouse and human.

Reply: There are unfortunately no commercially available antibodies that would distinguish murine and human Galectin-3. We have tested the antibodies used in our studies by Western blotting of mouse and human lysates separately and they clearly detect both (for example Figure S2D). However, and interestingly, mouse and human differ in the number of repeats present in the unstructured N-terminal end (Figure S2F). As a result, human Galectin-3 is shorter than mouse Galectin-3 and can be distinguished if using suitable SDS-PAA gel electrophoresis conditions: Figure S2C shows that OP9 cells with knockout of endogenous Galectin-3 but transduced with human Galectin-3 show a band that migrates faster than the mouse Galectin-3 [the human Galectin-3 also has an HA tag which makes it slightly larger]. Fig. S2D shows that human Galectin-3 made by human NKtert stromal cells similarly migrated somewhat faster than the Galectin-3 found in US7 cells co-cultured with wild type OP9 cells. Finally, to detect both human and mouse Galectin-3 in a co-culture system, we made use of the human chronic myelogenous leukemia cell line K562, which expresses endogenous Galectin-3 constitutively. When K562 cells are co-cultured with mouse embryonic fibroblasts, two Galectin-3 bands are seen: human and mouse. Thus, myeloid leukemia cells may be different than BCP-ALL in that they constitutively express Galectin-3 endogenously.

Reviewer 2 Report

The authors have presented an interesting piece of work investigating a role of Galectin 3 in the ALL.

The study is well designed and well presented. I just have one concern:

The authors did not measure the in vitro extracellular levels of Gal-3. Is there a reason for that? I strongly feel that measuring extracellular Gal3 (using an ELISA kit) from these given cells would add value to this paper. Currently, it is unclear if Gal3 production is exclusive to these cells in disease or other cells produce it too under the in vitro conditions studied in this paper.

I would also recommend using 2-3 housekeeping/reference proteins in the western blot.

Author Response

The authors have presented an interesting piece of work investigating a role of Galectin 3 in the ALL.

The study is well designed and well presented. I just have one concern:

Comment 1: The authors did not measure the in vitro extracellular levels of Gal-3. Is there a reason for that? I strongly feel that measuring extracellular Gal3 (using an ELISA kit) from these given cells would add value to this paper. Currently, it is unclear if Gal3 production is exclusive to these cells in disease or other cells produce it too under the in vitro conditions studied in this paper.

Reply: We thank the reviewer for the positive comments. Regarding ELISA measurements of Galectin-3, we agree with the reviewer that measuring Galectin-3 extracellular levels is important in this context. In a 2013 publication https://pubmed.ncbi.nlm.nih.gov/23760399/ we measured Galectin-3 levels secreted by OP9 cells into the medium (the data were shown as Supplemental data, not in the main manuscript). We showed that OP9 cells secrete up to 600 pg/ml of medium per 10^4 cells, and the presence or absence of human BCP-ALL cells only modestly enhances the Galectin-3 amounts. In addition, that study reported that BCP-ALL patient bone marrow plasma contains elevated levels of Galectin-3 compared to normal controls. We have added this information in the revised manuscript.

In our 2015 study using a co-culture of mouse Galectin-3 knockout MEFs and Bcr/Abl+ mouse BCP-ALL cells https://pubmed.ncbi.nlm.nih.gov/25869099/ (Figure 6) we concluded that under steady-state conditions [no chemotherapy] mouse BCP-ALL cells do not synthesize much if any endogenous Galectin-3 but that chemotherapy induces its endogenous production.

For further discussion as well as data regarding the issue if the BCP-ALL leukemia cells also produce Galectin-3, please see our response to Reviewer #1 and the new Supplementary Figure S2.

Comment 2: I would also recommend using 2-3 housekeeping/reference proteins in the western blot.

Response: We agree with the reviewer that it is important, depending on the experiment, to use additional methods of quantification. We routinely performed BCA assays to do this. In addition, we agree that actin is not always a dependable loading control. Depending on the treatment of the cells, levels of this housekeeping protein may also vary. For the Western blot shown in the figure we did not do additional immunoblotting, but the results in the figure are corroborated by many other Western blot experiments we performed with these cells and many other antibodies used on such Westerns. We here have added a multi-panel Supplementary Figure S2 in response to the current review in which we use Galectin-1, histone H3 and a-tubulin as other loading controls.

Round 2

Reviewer 1 Report

The authors responded to the concerns about statistical assessment and the origin of Gal3. Now the contents can be discussed.

Major

1. Authors replied that “In our study we did not perform quantitation on the FACS results for the different Galectin-3 levels, but we also did not make claims about the levels of Galectin-3 in the different OP9 cells that would require substantiation by such analysis.”

However, In the P7 L 260-261, authors claim about the levels of Gal3 mentioning that “Initially the sample had a signal for Gal3 that was higher than normal BM. After a 10-day co-culture with wild type OP9 cells, the signal was clearly increased, suggesting uptake of stromal-produced Ggal3 in these cells.”

To conclude their claim, substantiation would be needed with quantitation on the FACS results for the different Galectin-3 levels.

2. As authors mention in the manuscript, it is difficult to exclude the possibility of off-target effects. To conclude that “Gal3 is a valid target for enhancing the effects of standard chemotherapy by interfering with the communication between BCP-ALL and stromal cells.” (L66-68), the results of Fig7 and 8 should be compared with the coculture system on OP9-Gal3-KO same as Fig4.

Minor

1. In Figure S1, Ref #48 for GSE67684 should be “Br J Haematol 2018 Jun;181(5):653-663.”

2. In L 228-229 (Fig. S1C), it is difficult to conclude “increase” because there is no significant difference with this P value (0.0623).

Author Response

The authors responded to the concerns about statistical assessment and the origin of Gal3. Now the contents can be discussed.

Major

  1. Authors replied that “In our study we did not perform quantitation on the FACS results for the different Galectin-3 levels, but we also did not make claims about the levels of Galectin-3 in the different OP9 cells that would require substantiation by such analysis.” However, In the P7 L 260-261, authors claim about the levels of Gal3 mentioning that “Initially the sample had a signal for Gal3 that was higher than normal BM. After a 10-day co-culture with wild type OP9 cells, the signal was clearly increased, suggesting uptake of stromal-produced Ggal3 in these cells.” To conclude their claim, substantiation would be needed with quantitation on the FACS results for the different Galectin-3 levels.

Reply: The FACS plot shown in Fig. 2A was from an experiment that was performed five years ago, from 04/8/2016 to 04/28/2016 and was run on a C6 Accuri FACS machine. The results were only saved as .C6 extension files. BD no longer supports the C6 machines, and the one we have is no longer functional. I have tried to open the files with a free trail download of FlowJo v10.8.1 and free downloads of FCSalyzer and FCS Express but was not successful. Thus we are not able to provide MFI for these plots. We have removed the statement mentioned by the reviewer and modified the text. We have removed the former Fig. 2A from the main manuscript, include it in a new modified Supplementary Figure S2 and added FACS on five primary ALL samples showing a variable percentage of Gal3 positive cells.

  1. As authors mention in the manuscript, it is difficult to exclude the possibility of off-target effects. To conclude that “Gal3 is a valid target for enhancing the effects of standard chemotherapy by interfering with the communication between BCP-ALL and stromal cells.” (L66-68), the results of Fig7 and 8 should be compared with the coculture system on OP9-Gal3-KO same as Fig4.

Reply: We have added a section to the discussion comparing the results of Fig7/8 to those in Fig4 as indicated by the reviewer. To maintain a more logical organization of the text to accommodate this addition we re-organized and partly re-wrote the last part of the discussion.

Minor

  1. In Figure S1, Ref #48 for GSE67684 should be

Reply: We have corrected the reference and also cited it in the main text.

  1. In L 228-229 (Fig. S1C), it is difficult to conclude “increase” because there is no significant difference with this P value (0.0623).

Reply: We have deleted this sentence.

Reviewer 2 Report

Thanks for addressing the queries. 

Author Response

We thank the reviewer for taking the time to review our manuscript. 

Round 3

Reviewer 1 Report

In the revised paper, the authors  addressed all questions.